# Association between Prehospital Hypoxemia and Admission to Intensive Care Unit during the COVID-19 Pandemic: A Retrospective Cohort Study

**DOI:** 10.3390/medicina57121362

**Published:** 2021-12-14

**Authors:** Rémy Midez, Christophe A. Fehlmann, Christophe Marti, Robert Larribau, Frédéric Rouyer, Filippo Boroli, Laurent Suppan, Birgit Andrea Gartner

**Affiliations:** 1Division of Anesthesiology, Department of Anesthesiology, Clinical Pharmacology, Intensive Care and Emergency Medicine, University of Geneva Hospitals and Faculty of Medicine, 1205 Geneva, Switzerland; remy.midez@hcuge.ch; 2Division of Emergency Medicine, Department of Anesthesiology, Clinical Pharmacology, Intensive Care and Emergency Medicine, University of Geneva Hospitals and Faculty of Medicine, 1205 Geneva, Switzerland; christophe.fehlmann@hcuge.ch (C.A.F.); robert.larribau@hcuge.ch (R.L.); frederic.rouyer@hcuge.ch (F.R.); laurent.suppan@hcuge.ch (L.S.); 3Division of Internal Medicine, Department of Medicine, University of Geneva Hospitals and Faculty of Medicine, 1205 Geneva, Switzerland; christophe.marti@hcuge.ch; 4Division of Intensive Care Medicine, Department of Anesthesiology, Clinical Pharmacology, Intensive Care and Emergency Medicine, University of Geneva Hospitals and Faculty of Medicine, 1205 Geneva, Switzerland; filippo.boroli@hcuge.ch

**Keywords:** emergency medical services, prehospital, COVID-19, emergency department, intensive care unit, acute respiratory distress syndrome, peripheral oxygen saturation, hypoxemia, triage, orientation

## Abstract

*Background and Objectives*: The aim of this study was to assess the association between prehospital peripheral oxygen saturation (SpO_2_) and intensive care unit (ICU) admission in confirmed or suspected coronavirus disease 19 (COVID-19) patients. *Materials and Methods*: We carried out a retrospective cohort study on patients requiring prehospital intervention between 11 March 2020 and 4 May 2020. All adult patients in whom a diagnosis of COVID-19 pneumonia was suspected by the prehospital physician were included. Patients who presented a prehospital confounding respiratory diagnosis and those who were not eligible for ICU admission were excluded. The main exposure was “Low SpO_2_” defined as a value < 90%. The primary outcome was 48-h ICU admission. Secondary outcomes were 48-h mortality and 30-day mortality. We analyzed the association between low SpO_2_ and ICU admission or mortality with univariable and multivariable regression models. *Results*: A total of 145 patients were included. A total of 41 (28.3%) patients had a low prehospital SpO_2_ and 21 (14.5%) patients were admitted to the ICU during the first 48 h. Low SpO_2_ was associated with an increase in ICU admission (OR = 3.4, 95% CI = 1.2–10.0), which remained significant after adjusting for sex and age (aOR = 5.2, 95% CI = 1.8–15.4). Mortality was higher in low SpO_2_ patients at 48 h (OR = 7.1 95% CI 1.3–38.3) and at 30 days (OR = 3.9, 95% CI 1.4–10.7). *Conclusions*: In our physician-staffed prehospital system, first low prehospital SpO_2_ values were associated with a higher risk of ICU admission during the COVID-19 pandemic.

## 1. Introduction

In December 2019, a novel coronavirus (severe acute respiratory syndrome coronavirus 2 [SARS-CoV-2]) emerged in China and spread rapidly, leading to a global pandemic. The clinical manifestations of the associated disease, “coronavirus disease 19” (COVID-19), ranged from mild illness with fever and cough to severe pneumonia [1]. Clinical knowledge regarding COVID-19 complications was scarce at the beginning of the pandemic, the “acute respiratory distress syndrome” (ARDS) being most described [2]. Symptoms and signs of COVID-19 were, however, very heterogeneous, including in the prehospital setting [3].

Reports published by the first countries affected by the epidemic raised the awareness and anticipation necessary to limit the overload of health systems. As emergency departments (EDs) can easily be overwhelmed by a massive influx of patients [3,4], rapid triage methods are mandatory to refer patients to the most adapted care units in a timely manner and decrease the risk of overload. To avoid wasting valuable time and resources, patients requiring advanced respiratory care should be referred to intensive care units (ICUs) as soon as the need for such support is identified. Several international guidelines recommend that patients requiring endotracheal intubation (ETI) or presenting a peripheral oxygen saturation (SpO_2_) < 90% on oxygen (O_2_) with persistent signs of respiratory insufficiency should be admitted to an ICU [5]. Since the second wave and in the absence of immediate indications for invasive mechanical ventilation, high-flow nasal cannula oxygen (HFNC), noninvasive continuous positive airway pressure (CPAP), as well as prone positioning, have also been recommended in hospitalized patients, as they might decrease the need for intubation [6].

Even though most COVID-19 patients taken care of by prehospital providers do not require advanced respiratory care at first, about 5% of COVID-19 cases present a rapid worsening of their respiratory status and require invasive ventilatory support either in the prehospital setting or upon arrival at the hospital [7,8,9]. Early identification of these patients can help spare ED resources.

Few investigations can be performed in the prehospital field, but SpO_2_ is a simple tool available in all ambulances. Given the potentially elevated number of patients and the limitation of prehospital human and material resources, assessing the usefulness of this widespread measurement as a referral and triage tool was deemed of importance.

Our objective was to study the association between prehospital SpO_2_ and ICU admission in suspected or confirmed COVID-19 patients. We also wanted to assess the association between prehospital SpO_2_ and 48-h and 30-day mortality.

## 2. Materials and Methods

### 2.1. Study Design

This was a retrospective, single-center cohort study, approved by the regional Ethics Committee (Project ID 2020-01021).

### 2.2. Study Setting

The study took place in Geneva, Switzerland, whose detailed organization of prehospital emergency services has already been described [10]. Summarily, emergency calls are handled by professional dispatchers (paramedics or nurses), and the emergency prehospital response in Geneva is two tiered. The first tier is composed of an advanced life support ambulance staffed by two paramedics, which can be reinforced by way of a light vehicle (SMUR–Service Mobile d’Urgence et de Réanimation) staffed by a paramedic and an emergency physician. Per protocol, both an ambulance and a SMUR unit are dispatched whenever an acute respiratory distress is identified on call. The SMUR performs more than 5000 missions a year and belongs to the ED of the Geneva University Hospitals (HUG), a primary and tertiary care urban teaching hospital admitting 70,000 patients annually. In addition to the HUG, which were defined as the regional “COVID-19 Hospital” during the first wave of pandemic, there is one privately owned hospital and several clinics which were recruited to admit most non-COVID-19 patients. Prehospital patients for whom COVID-19 was considered the most likely diagnosis, based on respiratory symptoms and regardless of the presence of a fever, were treated with oxygen titrated for a target SpO_2_ > 90% or by invasive mechanical ventilatory support following ETI, if there were clinical markers of severity. Three such markers were defined: persistent respiratory insufficiency, coma and hemodynamic instability (Heart Rate (HR) < 40 or > 130/min, or systolic blood pressure (SBP) < 90 or > 180 mmHg). SpO_2_ values were measured with infrared digital pulse oximeters that meet the pre-analytical requirements for certification and comply with European Union (EU) certification standards. While this ensures that only high-quality monitors can be used in our system, different ambulance services have acquired different monitors, since the Geneva Directorate of Health has not issued specific brand guidelines regarding this material. In our system, patients are, as a rule, usually taken care of in the ED before being admitted to the ICU. Admission criteria to the ICU were determined based upon the Swiss Academy of Medical Sciences (SAMS) and institutional guidelines [4,11,12]. Their detail is described in Appendix A.

Non-invasive ventilation (NIV), which is part of our standard of care, was temporarily discontinued during the first months of the COVID-19 pandemic in an attempt to limit rescuers’ exposure to aerosolization hazards in the closed environment of the ambulance. The only exception to this rule was apyretic acute cardiogenic pulmonary edema, as NIV is particularly efficient in such situations [10] and as the probability of concomitant COVID-19 infection was considered as reasonably limited in this setting.

### 2.3. Selection of Participants

We proceeded to a computer screening of our database based on diagnostic codes specific to our prehospital unit. We included all patients aged 18 years or older with a prehospital diagnosis of suspected or confirmed COVID-19, dyspnea and pneumonia. All presumed diagnoses were made by prehospital physicians according to the presence of respiratory symptoms associated or not with fever [12].

Patient exclusion criteria were: a prehospital confounding respiratory diagnosis (such as acute pulmonary edema, pulmonary embolism), limitation of care (patients with a “do not resuscitate” or a “do not intubate” order or decision) clearly described in the prehospital file or in the emergency file and all files with a signed document indicating a refusal to participate in a clinical study.

We also decided not to include patients with a National Advisory Committee for Aeronautics prehospital severity score (NACA) of 7 (deceased on site), as well as patients for whom the paramedics had only requested medical advice by telephone. Patients transported to a private clinic were excluded as well, as only non-COVID-19 patients were accepted in these institutions during the first wave of pandemic.

Due to the retrospective nature of the study design, we used a convenience sample, which included all the patients treated by the SMUR who met the inclusion criteria.

### 2.4. Outcomes

The primary outcome was ICU admission during the first 48 h following the SMUR intervention. Secondary outcomes were 48-h and 30-day mortality. These data were manually retrieved from medical charts.

### 2.5. Variables

The primary exposure was the first prehospital SpO_2_ value measured upon arrival of the SMUR, before any medical treatment. This value is systematically collected and documented in the prehospital medical report. For the purpose of this study, we defined low SpO_2_ as a SpO_2_ < 90%. 

Other independent variables included: sex, age, intervention time (weekend, night), vital signs (HR, SBP, respiratory rate (RR), Glasgow Coma Scale (GCS) and NACA scores, which were collected for descriptive purposes.

Night interventions were defined as those occurring between 7 p.m. and 7 a.m. GCS was dichotomized using <15 as a cut-off to compare patients with a normal GCS to patients with an abnormal GCS. NACA scores were dichotomized in 2 categories of severity, using a usual cut-off linked to a higher estimated vital risk (NACA ≥ 4).

### 2.6. Data Collection

A computerized medical file with standardized fields is filled in by the SMUR physicians for every patient and reviewed by a senior physician as to ensure both data quality and teaching. We proceeded to an electronic screening of our prehospital database in order to extract all patient files recorded between 11 March and 4 May 2020. Out of these, eligible patient care files were manually identified by one of the authors (RM). Quality control of the selection of files was carried out by a second author (BG). Prehospital data were extracted to a Microsoft Excel^®^ (Microsoft Corporation, Redmond, WA, USA) and manually merged with data out of corresponding patient files of the ED, ICU and medical wards.

### 2.7. Statistical Analysis

Stata 16 (StataCorp LLC, College Station, TX, USA) was used for statistical analysis. Patient characteristics are described using frequency and proportion for categorical variables and means with standard deviation for continuous variables. We first compared two groups: patients with SpO_2_ < 90% (“Low SpO_2_”) with patients with SpO_2_ ≥ 90% (“Normal SpO_2_”), using the Chi-Square test or Student’s t-test, depending on the type of variables. Then, univariable logistic regression was performed to compute the crude odds ratios for the association between low SpO_2_ and each outcome. Finally, an exploratory multivariable logistic regression was performed for the primary outcome (respecting a ratio of 7–10 events per variable) and included two potential confounders (age and sex). These variables were chosen based on previous knowledge regarding ICU admission. As the assumption of the linearity of the log odds was not respected, age was dichotomized, using 65 as a cut-off. We also graphically represented the crude association between SpO_2_ and 48-h ICU admission using restricted cubic splines. A two-sided *p*-value < 0.05 was considered significant.

## 3. Results

A total of 890 patient files were screened. Inclusion criteria were met by 191 patients, of whom 46 were excluded according to our research protocol. We finally included 145 patients, all of whom were analyzed (Figure 1: Flowchart of inclusion). About half were male (55.5%), with an average age of 64.9 years. Interventions took place mostly on weekdays, during daytime (Table 1).

Mean SpO_2_ was 92.1%. A total of 41 (28.3%) patients had a low SpO_2_. Statistically significant differences between patients with normal and low SpO_2_ included age, respiratory rate and case severity (NACA score ≥ 4) (Table 1).

Among this population, 21 (14.5%) patients were admitted to the ICU during the first 48 h of their stay (Table 2).

There was a positive association between low SpO_2_ and ICU admission (OR = 3.4, 95% CI 1.2–10.0). Figure 2 shows the graphical representation of the crude association. After adjusting for sex and older age, the association remains statistically significant (aOR = 5.2, 95% CI = 1.8–15.4).

Regarding mortality, 7 (4.8%) and 18 (12.4%) patients died at 48 h and 30 days, respectively (Table 2). Low SpO_2_ was associated with mortality at 48 h (OR = 7.1 95% CI 1.3–38.3) and at 30 days (OR = 3.9, 95% CI 1.4–10.7).

## 4. Discussion

Our study highlights the fact that patients with a suspected or confirmed COVID-19 who present a first low prehospital SpO_2_ value are at higher risk of ICU admission and death. 

Previous studies have shown that a low prehospital SpO_2_ in COVID-19 patients was associated with intrahospital mortality, length of stay and need for prehospital ETI [13,14]. Some severely hypoxemic COVID-19 patients do not initially complain of dyspnea nor display proportional signs of respiratory distress, a phenomenon called “happy hypoxemia”. Indeed, in the initial phase of COVID-19, arterial hypoxemia is induced by intrapulmonary shunting and loss of lung perfusion regulation, without a concomitant increase in work of breathing [15]. Prehospital SpO_2_ values lower than 90% are, nevertheless, generally correlated with respiratory deterioration over time [2,15]. Such patients are therefore at increased risk of sudden worsening, requiring ETI and ICU admission [8,15]. Accordingly, we found that patients who belonged to the “Low SpO_2_” patient group had higher respiratory rates and higher prehospital gravity scores than those who belonged to the “Normal SpO_2_” group. While SpO_2_ should be interpreted with caution due to the leftward shift in the oxyhemoglobin dissociation curve linked to respiratory alkalosis, it remains a simple means often used to detect hypoxemia [15]. Given the prognostic importance of timely therapeutic and orientation decisions, our study supports the use of pulse oximetry as part of rapid assessment in the event of a massive influx of COVID-19 patients [8,14].

In the context of the uncertainties of the first wave of the pandemic and the possibility of setting-up triage scenarios at the hospital door, having such a simple and reliable orientation tool for prehospital health care staff could have provided a significant benefit. Indeed, the staff operate urgently, in an environment in which the logistical and primary care constraints are naturally complex. Wearing personal protective equipment (PPE), limiting aerosolization, shortages of medical devices and the large number of patients to take care of have all increased these constraints [16], making it difficult to consider easily the systematic use of invasive arterial blood gas measurements [7] or use of scales intended either for patient orientation in hospital departments [9] or to identify the need for hospitalization in outpatient setting. As an example, the OUTCoV score which includes symptoms, age and comorbidities, has a very good discrimination capacity but was not developed for emergency medical services (EMS) purposes. It is of interest to report that the authors plan to add SpO_2_ to this model as a supplementary risk factor in order to increase its capacity to rule out the risk of hospitalization among patients with SARS-CoV-2 infection in the outpatient setting [17]. Therefore, an initial normal SpO_2_ in the absence of risk criteria, such as older age, cardiovascular disease or immunosuppression, and in the absence of worrisome clinical signs, could possibly contribute to the decision to admit patients to general ward, or even to leave them at home under favorable conditions, thus sparing overcrowded intermediate and intensive care units [13,17]. Conversely, low SpO_2_ could be part of a decision-making protocol for direct admission to the ICU-bypassing ED [11,12,18].

The significant increase in the number of calls related to COVID-19 patients, as well as the strong demand for information during the first wave, put the EMS dispatch call centers under major pressure [19]. Home monitoring of stable patients had been set up with regular telephone assessments (partly video-assisted) in our local dispatch center, allowing early identification of clinical deterioration, as described elsewhere [20]. Both an ambulance and a SMUR unit are dispatched whenever an acute respiratory distress is identified on call. Nevertheless, in the context of an emerging disease, dreaded to worsen rapidly, ambulance and a SMUR were sometimes also dispatched in case of doubt concerning clinical gravity, which could in part explain why we found 72% of SpO_2_ in the range of normal values.

The positive association between prehospital SpO_2_ and admission to ICU is of interest in order to assess SpO_2_ relevance as a possible tool allowing early detection of patients to be referred from the prehospital setting to hospitals with the highest level of care, or even to consider fast tracks to ICU admission [11,13]. Obviously, prehospital triage allowing the best choice for orientation must be distinguished from triage performed on hospital admission. Indeed, eligibility for ICU admission cannot depend solely on SpO_2_ threshold and must be associated with as thorough a clinical and ethical assessment as possible. Many countries have developed specific triage guidelines and combination of criteria for COVID-19, including the assessment of vital functions, the probability of survival, comorbidities and the ethical principles [4,5]. Although we were able to adjust for age and sex in our analysis of ICU admission according to SpO_2_, adjustment on multiple age categories, as well as on a hemodynamic parameters and a frailty score [21] would possibly allow medics to determine more precisely the advantage of prehospital SpO_2_. Indeed, as described by Bavaro et al., older age, higher clinical frailty scale (CFS), as well as dehydration on admission were all independent predictors of mortality in a cohort of ≥65 years patients admitted to hospital with SARS CoV-2 infection. Moreover, the need for non-invasive and invasive ventilation was independently associated with mortality [22]. Concerning the positive association between low SpO_2_ and mortality, which is in accordance with previous publications [23], we were unfortunately not able to adjust our results given the low number events, which limit the scope of these results.

This study has several limitations. The first is that it was a retrospective study with a convenience sampling of a single facility. Given the limited sample size, we were unable to adjust for all confounding factors. Second, the very nature of prehospital emergency medicine whose diagnoses are based on clinical criteria may have induced selection bias. Third, although major prehospital alternative respiratory diagnosis were the exclusion criteria, we cannot exclude that some patients might have presented complications linked to COVID-19, such as sepsis, for example, which could have induced a concurrent cause of lowered SpO_2_ and ICU admission. Fourth, SpO_2_ is not the gold standard to reflect partial pressure of arterial oxygen (PaO_2_) and, in particular, SpO_2_ values in skin-pigmented people could be overestimated [24]. Furthermore, while EMS might intervene at any time of the disease course, patients are mostly hospitalized in acute care units between the 7th and 15th day after the onset of symptoms [1]. Unfortunately, we were unable to determine the median time between symptom onset and prehospital clinical assessment, as these variables were not systematically collected. It was therefore impossible to determine whether the lower SpO_2_ values correlated with a specific time since the onset of the disease, which could have been in turn correlated with ICU admission. Finally, the study began in the uncertainty of the epidemiological evolution, not allowing us to predict the generalizability of the results to the evolution of the pandemic and to the reorganization of the health systems which were to follow. As an example, the expansion of non-invasive respiratory support measures over the months generated changes in the organization of in-hospital orientation and eventually in ICUs [25].

## 5. Conclusions

Despite its limitations, this study shows that low prehospital SpO_2_ in patients with COVID 19 is associated with 48-h ICU admission, suggesting that prehospital SpO_2_ could be a useful tool to integrate to the triage of these patients in the perspective of future pandemic waves for early detection of those who require an ICU level of care. A multicentric prospective study encompassing subgroup analysis according to other clinical parameters should confirm our results.

## Figures and Tables

**Figure 1 medicina-57-01362-f001:**
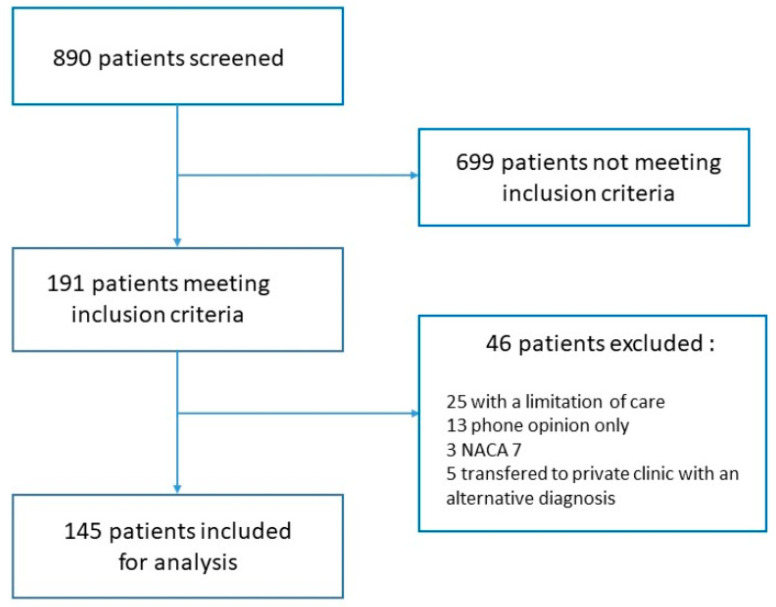
Flowchart of inclusion.

**Figure 2 medicina-57-01362-f002:**
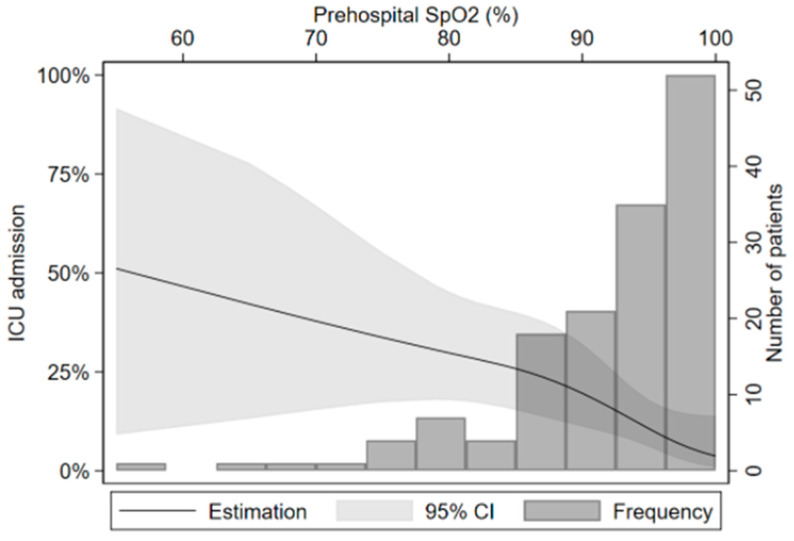
Association between oxygen saturation and ICU admission.

**Table 1 medicina-57-01362-t001:** Patient characteristics.

	Overall *(N* = 145*)*	Normal SpO_2_ *(N* = 104*)*	Low SpO_2_ *(N* = 41*)*	*p*-Value
Male—*n* (%)	79 (54.5)	55 (52.9)	24 (58.5)	0.538
Age (years)—mean ± SD	64.9 ± 17.4	61.6 ± 18.2	73.5 ± 11.4	<0.001
Age—*n* (%)				0.001
≤65	78 (53.8)	65 (62.5)	13 (31.7)	
>65	67 (46.2)	39 (37.5)	28 (68.3)	
Weekend—*n* (%)	33 (22.8)	21 (20.2)	12 (29.3)	0.240
Night—*n* (%)	52 (35.9)	37 (35.6)	15 (36.6)	0.909
HR (/min)—mean ± SD	97.7 ± 20.6	96.4 ± 20.1	100.9 ± 21.6	0.248
SBP (mmHg)—mean ± SD	137.1 ± 25.4	137.1 ± 22	137.1 ± 33	0.999
SpO_2_ (%)—mean ± SD	92.1 ± 7.8	96.0 ± 2.9	82.2 ± 7.5	NA
RR (/min)—mean ± SD	27.3 ± 10.8	25.2 ± 10.9	32.5 ± 8.9	<0.001
GCS—*n* (%)				0.441
<15	21 (14.5)	13 (12.5)	8 (19.5)	
15	106 (73.1)	79 (76.0)	27 (65.9)	
Missing	18 (12.4)	12 (11.5)	6 (14.6)	
NACA—*n* (%)				<0.001
<4	39 (26.9)	38 (36.5)	1 (2.4)	
≥4	106 (73.1)	66 (63.5)	40 (97.6)	

HR, heart rate; SBP, systolic blood pressure; RR, respiratory rate; SpO_2_, peripheral oxygen saturation; GCS, Glasgow Coma Scale; NACA, National Advisory Committee for Aeronautics prehospital severity score.

**Table 2 medicina-57-01362-t002:** Primary and secondary outcomes: group comparisons.

	Overall *(N* = 145*)*	Normal SpO_2_ *(N* = 104*)*	Low SpO_2_ *(N* = 41*)*	*p*-Value
48-h ICU admission—*n* (%)	21 (14.5)	10 (9.6)	11 (26.8)	0.008
48-h mortality—*n* (%)	7 (4.8)	2 (1.9)	5 (12.2)	0.009
30-day mortality—*n* (%)	18 (12.4)	8 (7.7)	10 (24.4)	0.006

## Data Availability

The data presented in this study are available on request from the corresponding author. The data are not publicly available due to ethical reasons.

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
