# Peer review of "Association between Prehospital Hypoxemia and Admission to Intensive Care Unit during the COVID-19 Pandemic: A Retrospective Cohort Study"

_medicina, 2021, doi:10.3390/medicina57121362_

Round 1
Reviewer 1 Report
In this study authors evaluated the risk of ICU admission of patients with early COVID-19 based on peripheral O2 saturation. Authors suggested the validity of this easy tool to quickly detect subjects at elevated risk of hospitalization in ICU.
Overall, the sample size is not particularly wide to draw definitive conclusions, however the work is interesting in the perspective of future pandemic waves to early detect patients who require a ICU care.
Several points should be implemented:
1) Please declare and discuss, if possible, the time from symptom onset to clinical evaluation.
In fact, a lower SpO2 at first evaluation may be representative of advanced disease, that is in turn an important predictor of ICU admission.
2) Any patient presented other complication at admission that may be a concurrent cause of low oxygen and ICU admission (eg. Heart failure, sepsis, etc).
3) In discussion, please include a more detailed discussion of risk factor for complications in course of COVID-19. For instance, it as been suggested that is not the age itself but more the frailty state of patients that increase the risk of complications. By citing this work (Bavaro DF, et al. Peculiar clinical presentation of COVID-19 and predictors of mortality in the elderly: A multicentre retrospective cohort study. Int J Infect Dis. 2021 Apr;105:709-715. doi: 10.1016/j.ijid.2021.03.021) I suggest discussing this point.
Reviewer 2 Report
The assessement of Spo2 is valuable and could prove useful generally. I recommend for publication if the authors could address the following points.
Line 56: whilst the guidelines say this, this is a controversial area based on consensus opinion, and it would be more nuanced to refer to the recent evidence that CPAP/NIV/Prone positioning etc reduces the need for intubation. Certainly, early intubation has been associated with increased mortality, as has late intubation.
Line 109 is too long. Please shorten
Could the authors mention if all the patients were caucasian/white? we know that pulse oximetry has some issues in black patients. how was spo2 measured? what device? what this standard across all the ambulances?
Line 239, SPo2 is not used solely as a criteria for admission to ICU. a careful holistic assessment is usually required. I think the authors should revise large parts of their article in that i think Sp02 should be used mostly as a triage tool, and perhaps those with non-worrisome Spo2 could be turned away home, but only after an assessment is done - pre-field hospitals etc
The authors say this is a retrospective study, then say this is a pilot study. It is not known what this is. A pilot would be prospective. And there is no clear control for confounders- that is a major limitation....
Round 2
Reviewer 2 Report
Dear authors, I am now happy with the changes, and the article as a whole. No further modification is required. Well done.